# Pig Farming in Alternative Systems: Strengths and Challenges in Terms of Animal Welfare, Biosecurity, Animal Health and Pork Safety

**Maxime Delsart [1],\*, Françoise Pol [2], Barbara Dufour [1], Nicolas Rose [2] and Christelle Fablet [2]**

[1]  Epidémiologie des Maladies Animales Infectieuses (EPIMAI), Ecole nationale vétérinaire d'Alfort, Unité sous contrat Agence nationale de sécurité sanitaire de l'alimentation, de l'environnement et du travail (ANSES), 94700 Maisons-Alfort, France; maxime.delsart@vet-alfort.fr (M.D.); barbara.dufour@vet-alfort.fr (B.D.)

[2]  Epidemiology Health and Welfare Research Unit, Ploufragan-Plouzané-Niort Laboratory, French Agency for Food, Environmental and Occupational Health and Safety (ANSES), 22440 Ploufragan, France; nicolas.rose@anses.fr (N.R.); françoise.pol@anses.fr (F.P.); christelle.fablet@anses.fr (C.F.)

\*  Correspondence: maxime.delsart@vet-alfort.fr; Tel.: +33-143-967-298

**Abstract:** In pig production, the widespread conventional indoor system with a slatted floor currently dominates. However, this production system is becoming less socially acceptable. In addition to general environmental protection issues, animal welfare, the absence of suffering and distress, and the management of pain also constitute societal concerns. In this context, alternative production systems are gaining ground. Although they are popular with consumers and other citizens, these alternative systems have their critical points. Here, we reviewed the international scientific literature to establish the state of the art of current knowledge regarding welfare, biosecurity, animal health and pork safety in this type of farming system. In general, alternative farms give pigs the opportunity to express a broader range of behaviours than conventional farms. However, the management of feeding, watering, temperature and predators is often more complicated in these outdoor systems. In addition, biosecurity measures seem to be applied less strictly in alternative farms than in conventional farms, especially in free-range systems, where they are more difficult to implement. On the other hand, pigs kept in these farming systems seem to be less affected by respiratory diseases, but parasitism and piglet crushing (in farrowing units) both remain a real challenge. Furthermore, the higher prevalence of many zoonotic pathogens in these farms may represent a risk for human health.

**Keywords:** outdoor; free-range; organic; welfare; biosecurity; health; safety

## 1. Introduction

Pork is the second-most consumed meat in the world [1]. Pig production has a very wide range of rearing methods [2,3], with conventional pig production in closed buildings with slatted floors—the currently dominant system—coexisting with other so-called alternative production systems in terms of housing. The demand for pork products in the future may change and be strongly influenced by socio-economic factors, including animal health concerns, but also changing socio-cultural values [4]. Increasing protests in the media and various actions by animal-welfare associations indicate that the current dominant production system is becoming less acceptable, especially with regard to animal welfare [5]. Numerous surveys throughout the world testify to this societal evolution. Whether in Brazil [6], the United States [7], Canada [8] or Europe [9], citizens are expressing their preferences for free-range animals, with no restrictions on movement. Animal welfare, the absence of suffering or

distress and pain management are concerns [6], as is environmental protection [9]. In both the United States and Europe, the "natural" aspect of farming is also a factor to better take into account in animal husbandry, with pigs having access to the outdoors [7]. In France, 60% of consumers consider it a priority to provide outdoor access for all animals [10]. In an American survey [7], 44% of respondents had concerns about the space provided for animals, with 20% believing it was necessary for animals to have outdoor access. According to Norwood and Lusk [11], consumers in three different U.S. locations (Chicago, IL; Dallas, TX; and Wilmington, NC) would be willing to pay $2.02 more per kg of pork chops raised in a pasture system as opposed to an indoor system.

It is in this context and with the growing markets for pork raised in more "natural" conditions or according to "organic farming" specifications, that alternatives to so-called "conventional" farms are being developed. We hereafter define as "alternative" as any farming system different from the predominant contemporary structures [12], i.e., not raising all pigs in closed buildings and on slatted and/or concrete floors. As with conventional farming, there is a wide variety of alternative production systems, which are generally more oriented towards animal welfare and quality [3]. There are both free-range and litter-based systems. Outdoor pig production is defined as a system that allows pigs to have outdoor access and be in contact with the ground and growing plants [13]. This system has expanded rapidly in parts of Europe, South Africa and North America [14], as well as in other parts of the world (e.g., in 2007, more than 60% of pigs were kept outdoors in Uruguay [15]). The number of pigs in these systems varies widely, from less than 10 to more than 10,000 sows [14]; on a given farm, all or only some of the pigs may have access to the outdoors (e.g., breeding stock or growing pigs [16]), while the rest of the pigs may be kept on slats or bedding. Finally, on some farms, outdoor access may be reduced to an outdoor yard open to the barn. Bedded housing has the same type of diversity [17], because not all pigs are necessarily raised on bedding within a given farm. There is also great diversity in the litter used (straw, sawdust, hay, etc.).

Although alternative breeding systems have been developed, they are currently not very attractive to farmers; for example, raising pigs on litter represents only 5% of pig farms in France [18]. Numerical data on alternative breeding systems are scarce. For instance, for organic livestock farming, which does not include all alternative livestock farms, statistics on the number of animals raised throughout the world according to "organic farming" specifications are incomplete and do not provide a complete picture of this sector for the moment. However, available data indicated that in European countries, 9 million organic pigs are produced per year, with a growth of 46% between 2007 and 2015 [19]. Nevertheless, in 2015, organic pig production represented only 0.5% of the total pig production in Europe. Again, there is a great diversity of organic farming systems within and between countries [20]. Outdoor rearing predominates at all physiological stages in Italy and Sweden and for sows in France and Denmark, but many organic pigs are kept indoors in Germany and Austria [21]. Housing may be diverse within the same farm, e.g., early gestation sows housed indoors and late gestation sows housed outdoors, as in France and Denmark. Italy differs strongly from other countries with smaller farms using local breeds [21]. The diversity of production systems in sustainable pig sectors has also been described as part of the TREASURE project [22].

Although they are popular with consumers and other citizens, these alternative farms have their critical points. The objective of this review was to establish the state of the art of current knowledge concerning welfare, biosecurity, animal health and pork safety in this type of farming system. The economic aspects will not be considered here, as the study of the economic motivations of farmers for the choice of farming systems and economic profitability of such systems are a research area per se. The bibliographic search engines PubMed and Google Scholar were queried, using the key words "pig" or "swine" associated with key words concerning the type of farming ("outdoor", "free range", "organic", "straw", "litter"), welfare ("welfare"), biosecurity ("biosecurity"), animal health ("health", "parasitism"), and food safety ("*Salmonella*", "*Toxoplasma*", "hepatitis", "antibiotic", "*Campylobacter*", "*Yersinia*", "*Trichinella*" and "*Listeria*"). The results of this research resulted in 2045 articles, 1360 of which were published after 2010 (66.5%). Several references from the authors' personal archives were also used. After reading the titles and abstracts, about 350 were read and 222 were finally selected for this review.

## 2. Animal Welfare and Alternative Production Systems

The average citizen does not know how livestock are raised. They often idealise alternative, farm-based farming, in which the animals have access to an outside run or straw area, rather than conventional farming on slatted floors [23]. One of the expectations of the consumer who buys organic food is that animal welfare standards are higher in these farming systems [23]. However, the concept of animal welfare is complex. Animal welfare refers to the psychological state of an individual in relation to its internal and external environment [24]. According to the French Agency for Food, Environmental and Occupational Health & Safety (ANSES), "animal welfare is the positive mental and physical state related to the satisfaction of its physiological and behavioural needs and expectations. This state varies according to the animal's perception of the situation" [25]. In this section, we summarise current knowledge on animal welfare in alternative pig production systems following the five freedoms approach established by the British Farm Animal Welfare Council (FAWC) [26].

### 2.1. Absence of Hunger and Thirst

In the wild, pigs are active during the day and spend 75% of their active time on foraging activities, including rummaging, grazing and exploring with their snout [27,28]. On the farm, the distribution of time is different, and pigs spend less time looking for food. Feed is distributed to the pigs in different forms (meal, pellets, liquid feed), in unlimited quantities or in limited quantities at specific times determined by the farmer. There are few differences in the way feed is distributed in conventional and alternative livestock production; the causes of stress due to feed deficiencies are the same, i.e., insufficient trough length or an insufficient amount of feed distributed to the animals [29]. However, for an equivalent growth performance, animals raised outdoors consume on average more feed than animals raised indoors [15], due to greater activity [30] and increased energy expenditure due to lower ambient temperatures, particularly in winter [31]. For the same amount of feed, the needs of a pig raised outdoors may not be covered and may generate physiological and behavioural stresses. However, ingestion of herbage and soil by pigs at pasture can make a non-negligible contribution to the energy, amino acid, mineral and micronutrient requirements, especially for sows with a high capacity to ingest bulky feeds [30]. The volume ingested depends very much on the nature of the forage and its palatability [32]. Overall, the palatability of the feed is important for feed intake and for the pig's ability to cover its needs. Beyond the presentation of the feed—the pig prefers pellets to meal [33]—a microbial contamination (e.g., a high level of colony forming units of yeasts), mycotoxins (especially vomitoxine) [34] or inadequate levels of specific amino acids (e.g., tryptophan [35]) can cause a decrease of feed intake. The following parameters require attention in alternative farming systems: (i) Feed preservation, especially when it is distributed outdoors; (ii) the amino acid balance, especially in organic farms where the incorporation of synthetic amino acids is not allowed; (iii) or the presence of mycotoxins which seems to be higher in organic wheat than in conventional wheat [36].

To objectify the absence of hunger, indicators can be used, such as body condition scores or the rate of lean animals [37]. A survey of 101 European organic farms showed that 18.8% of the sows were lean with high variability among farms [38]. This rate is significant compared to the rates reported in a study on 82 English or Dutch farms of all types, where only 0.1% of the sows were very thin, with no apparent difference between free-range and other sows [39]. These results suggest that some organic farmers have difficulties in meeting the nutritional requirements of sows. Another study showed that growing pigs raised on straw or extensive outdoor systems had a higher risk of poor body condition than pigs raised in conventional systems [40], a risk run particularly in production systems where pigs are entirely dependent on pasture.

The absence of thirst is ensured by providing a quantity of water to cover the animals' needs. These needs vary according to physiological stage, but also according to the environmental conditions [41]. Only permanent access to drinking water can meet the physiological needs of each pig at all times, depending on risk situations such as heat or disease [42]. The availability of drinking

water can be a problem, especially in extensive systems. Under free-range system, water troughs are often accessible to wild birds and contaminated with dust [43]. Access to poor quality water can be a cause of poor water intake and may induce health problems in animals [44]. Water troughs must be cleaned daily and must be equipped with a purge [16]. Water temperature control can also be a challenge. A high temperature may cause a low water intake as found in pigs housed in conventional confined settings [45]. Even though scientific literature on this subject is lacking, field experiences of the first author indicate that in outdoor systems, water supply pipes should preferably be buried to limit the effects of frost (water is no longer distributed to the animals in winter) and the action of the sun on the pipes promotes bacterial proliferation and a significant increase in water temperature which reduces the animals' consumption of water [16]. In addition, unprotected, unburied pipes are accessible to pigs, which can then destroy them [46].

## 2.2. Absence of Discomfort

The second freedom is the absence of discomfort, which is guaranteed by an appropriate environment, with sufficient space for the animals to move freely, comfortable draft-free resting areas, enough light for the pigs to be able to see and have a nychthemeral rhythm, and the necessary thermal comfort. Bedding made of straw has properties similar to the type of substrate a pig would naturally find, acting as a cushion and reducing discomfort and injury [47]. Several studies have shown that the risk of bursitis (a fluid-filled sac surrounded by fibroblasts that forms in subcutaneous connective tissue as a result of pressure exerted on the skin over a bony prominence [48]), is significantly lower in alternative farms with access to the outdoors or on straw than in conventional confined farms [49–51]. The severity of bursitis is associated with hard, uncomfortable environments that increase the pressure on joints. The rate and severity of bursitis tend to decrease during a straw or outdoor fattening phase, but increase when pigs are fattened on slatted floors [52].

Although resting time does not appear to be different for pigs reared on slatted floor or on straw [53], animals reared outdoors appear to be more active than animals reared indoors. In a Uruguayan study [15], a comparison was made between the behaviour of 48 pigs reared with an outdoor access and 48 pigs reared indoors. Animals in outdoor systems are in general more active and show a daily pattern of behaviour with two peaks of activity—in the morning and afternoon—whereas confined pigs are more sedentary and have a more stable behavioural pattern along the day. In another study [54] it was shown that all 86 Iberian pigs with outdoor access regrouped to rest during the night in a common area in the shelter. Beyond comfort, the shelter could be considered, according to the authors, as a place of refuge or hide from predators. In these shelters, animals tend to prefer place without air draught [55].

Pigs have limited thermal regulation capabilities; in the wild, they depend on mud baths to cool themselves on hot days [24]. In a thermoneutral situation, pigs separate their resting area from their manure area, which is preferably moist and exposed to draughts [56]. However, this dissociation disappears when pigs experience high temperatures and the entire pen is soiled with excrement [24]. Pigs respond to high ambient temperatures by using a floor that promotes conductive heat loss. Ducreux et al. [57] showed that pigs preferentially rest on litter at 18 °C and on concrete or slatted floors at 27 °C. However, apart from high temperatures, straw is much more comfortable for pigs than bare concrete floors [58]. Straw provides thermal comfort and can reduce the room temperature requirements of growing pigs by up to 6 °C [47].

For outdoor pigs, the main challenge is to keep them clean and dry in wet weather conditions [41]. The type and management of housing huts must be adapted. They should be comfortable and large enough to accommodate the pigs [16]. In farrowing paddocks, there is a seasonal effect on outdoor piglet mortality [59,60], related to the management of comfort during farrowing and the ability of the sow to prepare her nest correctly in the hut. Low density of ground cover in the pen, accompanied by the permanent presence of mud, increases humidity and discomfort in huts [41,59]. Piglets are very sensitive to cold and draughts at birth. Low ambient temperatures increase the proximity of piglets to their dam and favours mortality by crushing [41]. Crushing can be reduced by abundant mulching in huts [59]. Dry bedding provides a nest and thermal insulation, promoting

piglet survival. The choice of hut for outdoor rearing also affects the pre-weaning mortality rate, with real differences depending on the type of hut [14]. Those with the lowest pre-weaning mortality rate provide a large area where piglets can protect themselves from the sow. However, piglets can also suffer when it is very hot. The heat affects the sow's milk qualities, and the sow also spends more time outside the hut cooling down rather than nursing its piglets [60]. Limiting the effects of heat when pigs are reared outdoors requires providing them with shade or cooling areas, such as spray nozzles or wallows [16,61]. Wallows are also used to cool down in hot weather and to protect against insects [28].

As with the absence of hunger and thirst, animal indicators can be used to assess the presence of discomfort [37]. Determining rates of huddling, shivering and even panting may be of interest, but these indicators depend on factors other than the type of farming itself, such as weather conditions on the day of observation or animal density [40]. The presence of soiled pigs is an indicator of a dirty pen and therefore uncomfortable lying positions for the pigs; thus the cleanliness of the pigs may offer the possibility to assess the absence of discomfort [37]. For example, once the upper critical temperature has been reached, pigs raised on straw tend to cool down by wallowing in their own droppings [40]. However, assessing animal cleanliness is not so simple. Dippel et al. [38] pointed out a large variability in this criterion, linked to observers who do not differentiate between mud and manure.

### 2.3. Absence of Pain, Injury and Illness

The absence of disease will be developed below in Section 4, as will neonatal crushing mortalities which will be developed in Section 4.1 on the mortality of suckling piglets. Injuries and pain may be the result of fighting between animals, or of mutilations imposed by farming systems or farmers.

In the study by Dippel et al. [38], 15.5% and 7.9% of all sows had injuries from fighting, depending on whether the injuries were on the front or back side of the sow. However, these figures hide the great variability between farms, regardless of the farming system. Another study showed that sows fed in groups outdoors tend to have fewer body injuries or vulvar injuries than sows fed with electronic sow feeding (ESF) systems indoors [39]. However, the feeding pattern in that study seems to have had a greater impact on the difference in animal behaviour because sows fed indoors in groups did not show more injuries than sows fed outdoors [39].

Piglets kept outdoors show less agonistic behaviour towards other piglets than piglets kept indoors, either before or after weaning [62]. In a study of 1928 pig farms in the UK, there were significantly fewer severe injuries in growing pigs kept outdoors compared with confined pigs (0.12% (±0.84) and 0.29% (±1.96) respectively) [63]. Paradoxically, there were more pigs with lesions in farms in which indoor animals had outdoor access (0.33% ± 2.05), and the difference to pigs kept outdoors was significant. Equivalent observations were found for tail biting and cannibalism lesions [64]. Tail biting is a behavioural problem with multifactorial causes, including overcrowding [65], disease [66], feeding problems (e.g., competition for feed and/or inadequate feed intake, low fibre content, protein deficiency or imbalance, mineral imbalances), but also poor ventilation or uncomfortable temperatures [67,68]. Pigs kept in housing with outdoor access probably have more difficulties controlling their thermal environment when moving from a protected system to an outdoor system, which may explain that the agonistic behaviour is higher in this type of farming than in fully confined housing [63]. Similarly, studies conducted in slaughterhouses [50,51] have shown that pigs raised in outdoor runs have more tail injuries than pigs raised in conventional systems. However, tail docking is less often practiced in this type of production, in contrast to conventional production. This difference in practice may explain, at least in part, these observations at the slaughterhouse.

In contrast, tail biting is less frequent when pigs have straw [41], because the pigs spend more time examining the ground and moving around, and significantly less time biting the tails of other pigs compared to pigs raised on slatted floors [52]. Nevertheless, these data suggest that additional space, access to outdoor areas and the provision of straw are not sufficient to prevent whole-tailed pigs from biting their tails [50,69], the risk being higher in winter than in summer [50].

Injuries and pain can be induced by the production system itself and by the farmer. The main example, of course, is the surgical castration of pigs, which is still very common and is generally done without full pain management. Male piglets are castrated to limit sexual odours in meat and to avoid undesirable behaviour such as overlapping. Although surgical castration without anaesthesia is currently allowed at up to 7 days of age in Europe, it is often performed later in organic farming. A study published in 2012 reported that less than 10% of organic farmers castrate piglets without anaesthesia beyond the first week of life in Austria, Denmark, Germany and Sweden, but about 75% in France and Italy [21]. This late castration is due to greater organisational difficulties, especially in the outdoor systems [16,21].

Teeth clipping is rare in organic production [70] and outdoors. Its main objectives are to limit lesions on sows' udders and skin lesions on piglets, especially on the head. According to one study, the absence of teeth clipping on piglets raised outdoors has no impact on sow udders [71]. However, in the absence of teeth clipping, there is more skin damage on piglets, but without impacting piglet survival, health or weight gain [71].

On the other hand, nose-ringing in free-range pigs is a centuries-old practice still used today [28], mainly on sows, and especially in France, Denmark and Germany [70]. The active foraging behaviour of pigs can lead to the destruction of pastures [28,72], but this behaviour becomes painful with a ring in the snout. This pain and the pain caused by the ring, together with inhibition of rooting activity, are factors affecting the welfare of these free-range pigs [72].

*2.4. Freedom to Express Normal Behaviour*

Among the most frequent criticisms of conventional production systems is the lack of freedom for animals to express their natural behaviour [73]. However, the absence of natural behaviour does not necessarily imply suffering [28], but can be a source of frustration. Frustration can be expressed by higher levels of stress hormones [74], stereotypies, abnormal agitation levels, reduced play behaviour, redirected behaviour, especially towards other animals, and increased agonistic interactions that can lead to cannibalism [75].

As mentioned above, in the wild, pigs spend 75% of their active time exploring their environment [27,28] by searching, sniffing, biting and chewing consumable, but also non-digestible items [76]. Pigs are gregarious animals. They form stable groups with a hierarchical social structure, limiting severe physical aggression between individuals [24].

Many studies show that animals are more active in alternative production systems than in conventional systems, particularly for growing pigs reared outdoors [15] or on straw [52,77]. Pigs are more often standing up [15,23,52,77], whereas pigs in confinement are inactive most of the time [15,52]. On straw, growing pigs spend more time interacting with their environment, being exploratory [23,52,77]. However, there is high variability among farms [23].

Straw also reduces harmful social behaviour, such as ear or tail biting [78,79]. When it is not possible to provide straw, enrichment items can be provided [79,80]. However, many studies show that straw or other organic substrates used as bedding remain the most effective material to reduce inappropriate behaviour [40,63,81]. Straw therefore constitutes suitable environmental enrichment for pigs, allowing them to express their exploratory behaviour.

Pigs raised outdoors tend to exhibit less aggressive behaviour than pigs raised indoors, with fewer fights and mutual aggression [15]. This is particularly the case for suckling piglets, which spend less time interacting with their mothers and have less agonistic behaviour when outdoors [62].They spend more time exploring, feeding, walking and playing [62,82]. They benefit from ample space and enrichment of their environment that allows them to express their natural behaviour under good conditions [83]. Another factor that may induce different behavioural responses in piglets reared in conventional or free-range systems is the opportunity of having social interactions between piglets from different litters during lactation [84]. Piglets reared in group lactation are less aggressive after being mixed into groups with unfamiliar piglets at weaning than piglets reared in farrowing crates [85]. Their mothers also behave differently outdoors: They spend more time standing up exploring

the environment than confined sows [62]. Outdoor sows and piglets show a richer behavioural repertoire (ethogram) [82].

Straw and materials found outdoors allow sows to build a nest and can influence their maternal behaviour. In the wild, sows separate from the group a few days before farrowing and prepare a nest by digging the ground with their snout and making a series of trips to collect long grasses, leaves and small twigs [24,28], which can be easily replaced by straw.

*2.5. Absence of Fear, Stress and Anxiety*

Work on human–animal relationships describes an aptitude of pigs to develop fear and anxiety reactions when approaching humans [86]. This fear, which can be assessed specifically by the extent of animals' withdrawal behaviour in contact with humans [37], seems to be similar depending on whether sows are housed in groups indoors or outdoors [39]. Domestication of animals is equally important in alternative and confinement farming and farmers should be trained to manage and handle animals properly to reduce anxiety [87].

The cumulative effects of a sudden separation from the sow, a transition from a milk to a grain-based diet, mixing with unfamiliar pigs for the first time and a change of the physical environment make weaning a critical period of adaptation and stress, imposing high demands for adaptive processes [88,89]. Weaning appears to be less stressful for piglets in outdoor than in confined systems [62]. In a study comparing the behaviour of piglets weaned at 3 weeks according to their rearing system, piglets spent more time belly-nosing and displaying agonistic and oral–nasal behaviours directed to penmates than outdoor piglets, at weaning and after weaning [62]. In a Japanese study, two groups of piglets born and raised outdoors or in sized standard pen in a piggery were weaned at 4 weeks of age. The ears of all piglets were pierced at weaning using self-piercing ear tags, as a stress stimulus. After the stress treatment, salivary cortisol levels increased to a significantly lower level in the outdoor-housed group than in the indoor-housed group [90].

In natural or semi-natural conditions, piglet weaning is a progressive process characterised by a substitution of maternal milk by other food, starting around the 4th week of age and terminating between the 9th and 17th week of age [91,92]. Age at weaning influences the adaptation of outdoor-reared piglets to weaning. Increasing the age at weaning from 3 to 4 weeks improves the welfare of piglets reared outdoors by reducing distress behaviours and improving feeding behaviour [93]. On organic farms, piglets are weaned older, after 6 weeks of age. Several studies show that increasing the age at weaning can have a positive impact on salivary or plasma cortisol responses at weaning [89,94]. In a study with 160 piglets of 16 litters, plasma cortisol response at the day of weaning was lower in piglets weaned at 7 weeks compared with piglets weaned at 4 weeks [89].

Stress, fear and anxiety can also be induced by the presence of predators. Young piglets reared outdoors are exposed to predation, including corvids, foxes and even badgers [83,95]. A study analysing the causes of piglet mortality in free-range piglets [96] showed that 6% of piglet corpses showed evidence of bird attacks. There are prevention systems against some of these predators that can be set up if the pigs are exposed [41].

Summary (Table 1)

Alternative production systems have advantages and disadvantages in terms of animal welfare. Welfare is primarily linked to the characteristics of the environment. Extensive systems give pigs the opportunity to express a wider behavioural repertoire than indoors, including most of their natural behaviours, but controlling the environment is more difficult, including the management of feeding, watering, temperature and predators. Bedded systems also allow the expression of a wider ethogram and offer comfort for the pigs, provided the bedding is healthy and dry. Alternative systems, with an enriched environment provide ample solutions to control animal welfare critical points, unlike conventional farming, where pigs have more difficulties to express their natural behaviour, despite the addition of enrichment materials or even an increase in surface area per animal. Even in extensive outdoor systems, good husbandry and management must ensure animal welfare. Critical points

being under control, alternative systems are more suited to satisfy societal expectations in terms of animal welfare.

**Table 1.** Summary of the strengths and weaknesses in terms of animal welfare according to production method.

| | Conventional | Indoor with litter | Outdoor | Indoor with outdoor access | |
|---|---|---|---|---|---|
| Absence of hunger | +++ | ++ | + | ++ | [34–36,38–40] |
| Absence of thirst | +++ | +++ | + | +++ | [16,43] |
| Absence of discomfort: | | | | | |
| - Thermal: | | | | | |
| • Hot temperatures | ++ | ++ | ? | +++ | [24,28,40,57,60] |
| • Cold temperatures | +++ | +++ | ? | +++ | [41,47,58] |
| • Humidity | +++ | +++ | - | ++ | [41,59] |
| - Comfortable floor | - | +++ | +++ | +++ | [49–52] |
| - Comfortable resting area | + | ++ | ? | +++ | [14,15,24,53–56] |
| No pain, no injuries | - | +++ | ++ | + | [21,28,38,39,41,50–52,62,64,70–72] |
| Express normal behaviour | - | +++ | +++ | +++ | [15,23,40,52,62,63,77–79,81–85] |
| No stress, no fear | +++ | +++ | + | ++ | [39,62,83,89,90,93–95] |

+++ Very favourable; ++ fairly favourable; + not very favourable; - unfavourable; ? unknown.

## 3. Biosecurity and Alternative Breeding Systems

The concept of biosecurity encompasses the full range of measures employed to prevent the introduction and spread of diseases [97]. Several studies have established a correlation between the level of biosecurity and the technical performance of livestock production, such as growth or feed conversion rates [98,99]. Biosecurity can also help reduce the use of antimicrobials in animal husbandry, at least prophylactically [98].

Biosecurity measures are generally best applied in large herds, in modern facilities and by young farmers [98]. A study on 471 farms in the Philippines found that compliance with biosecurity measures was better on commercial farms than on small family farms, which in that study included almost all free-range farms [100].

Biosecurity encompasses all measures that need to be set up to limit the risk of introducing pathogens into the farm (bio-exclusion), to limit the spread of the pathogen within the farm (bio-compartmentalisation), to limit the spread of the infectious agent outside the farm (bio-containment), to prevent the risk of contamination of humans and to prevent environmental contamination and the persistence of the pathogen in the environment [101].

### 3.1. Bio-Exclusion

The pig industry regularly suffers from the devastating effects of infectious diseases. In the last 30 years, many diseases have emerged or re-emerged in the world, such as the porcine reproductive and respiratory syndrome (1991), porcine circovirus type 2 infections (1994), porcine epidemic diarrhoea (2013) or African swine fever in Europe and Asia (2007) [102]. Many are highly contagious viral diseases for which only sanitary prophylactic measures are effective [97]. Even long-standing enzootic diseases, such as enzootic pneumonia or swine dysentery, still cause heavy losses if introduced into naive herds [97]. Biosecurity measures aim to prevent the pathogens associated with these diseases from entering and contaminating the farm. Strict biosecurity measures are more difficult to implement in alternative farms with external access for the animals. The likelihood of exposure of these farms to several pathogens circulating in wildlife, such as the African swine fever virus, for example, is much higher [103,104]. This likelihood is especially high in very extensive systems, such as the widespread silvopastoral system in some southern European countries (e.g., Spain), where pigs graze in parts of natural forests [105]. Biosecurity is almost impossible to apply

when pigs have access to pastures shared by different herds, as is the case in Sardinia, Corsica or in public forests in Georgia or Armenia [106].

It is of course contact with wildlife, mainly wild boars, that constitutes the main issue when pigs are reared outdoors. Contact with wildlife is considered to be a source of infection, e.g., brucellosis, classical swine fever [107], African swine fever [103] or Aujeszky's disease [108]. The increasing numbers of wild boars in some regions, particularly in Europe, and their ability to colonise new areas, poses a threat to free-range farming [109]. A study carried out in Switzerland between 2008 and 2010 showed that contacts between wild boars and pigs kept outdoors are not uncommon and even lead to the emergence of pig-boar hybrids [110]. The greater the distance between the pen and the farm, the closer wild boars get to the fence where the pigs are raised (<2 m); the risk of intrusion increases if this distance is greater than 500 m and if the pen is protected only by a simple electric fence, or any other fence less than 60 cm high.

Pigs may also be in contact with other disease-vector species. Pigs kept outdoors are inevitably exposed to certain leptospiral serovars from a variety of wild species including hedgehogs, foxes and rats [97,109]. Rodents can be reservoirs for multiple pathogens that can affect pigs such as *Erysipelothrix rhusiopathiae* [111], *Brachyspira hyodysenteriae*, *Lawsonia intracellularis* [109], several species of *Salmonella* and *Yersinia*, protozoa such as *Toxoplasma gondii*, helminths such as *Trichinella* spp. and *Echinococcus multilocularis* [111]. The control of rats in alternative breeding systems is particularly complicated, as there are more shelters or nesting sites for rats and chemical control is not easy to implement. A study carried out in Denmark in 428 organic and free-range farms revealed a positive correlation between the presence of rats and straw stacks near the pig sties [111].

Birds are another vector that is very difficult to control in alternative systems. They may be involved, for example, in the spread of transmissible gastroenteritis [109], salmonellosis [109,112] or avian tuberculosis [109].

Other vectors include insects and ticks, such as *Ornithodoros moubata* and *Ornithodoros erraticus*, which have contributed to the transmission of African swine fever virus in outdoor production systems in the Iberian Peninsula [113,114].

Finally, animals raised outdoors are more accessible to curious visitors, which facilitates the transmission of zoonotic agents [105] or the distribution of leftover food contaminated with infectious agents such as African swine fever or classical swine fever viruses.

Another biosafety hazard is litter and in particular straw used in alternative animal husbandry. Several studies have shown that production systems including sawdust, wood shavings or straw increases the incidence of tuberculosis caused by *Mycobacterium avium* [115,116]. The bedding may have been exposed to other animals, and may have been contaminated [109]. Therefore, bedding should be protected and stored to prevent contact with domestic suids other than those kept on the farm, or with wild suids. It must also be protected from possible contamination by pests [117]. The same recommendations apply to feed distributed to animals [106,117].

To reduce contact between wildlife and domestic pigs reared with outdoor access, fencing is strongly recommended, if not mandatory [117]. The succession of health crises, and in particular the spread of African swine fever (ASF), has prompted the pork industry, institutions and governments to become aware of the importance of biosecurity. Numerous technical guidelines have been written [118–120]. Spain is an example of a country with strict biosecurity standards for outdoor pig production. Regulations regarding biosecurity on outdoor pig farms [121] are a result of the presence of ASF between 1960 and 1995 in the Iberian Peninsula. Another example is the French Ministry of Agriculture and Food who signed an order in 2018 regarding biosecurity measures applicable to farms holding swine [122]. The most important measures for alternative farms concern fencing to prohibit the intrusion of wild suids into the farming area and nose-to-nose contacts between farmed and wild suids. These measures apply to all types of production systems, not just outdoor systems, because this type of contact is also possible in farms with outdoor runs, or for animals kept in buildings with one or more sections of wall with openwork on the outside [117]. Fences are often expensive and difficult to install [95], but also difficult to maintain in the long term [106]. Their maintenance is essential, especially in the case of electrified systems. Any contact between the wires

and the ground reduces the efficiency of the fence (current leakage to the ground). Weed control around the fence is therefore necessary, but can quickly become time-consuming, especially in organic farms where the use of synthetic weed killers is generally prohibited.

### 3.2. Bio-Compartmentalisation

Bio-compartmentalisation involves limiting the spread of the pathogen within the farm, e.g., by adopting zone-specific clothing or isolating shedding animals, which may be done significantly less often in small, predominantly free-range farms than in almost exclusively indoor commercial farms [100].

An infection in a population can also be maintained in alternative farms through the presence of bedding that favours contact between the animals and their droppings. In contrast to conventional slatted floor systems, the types of flooring found in these alternative systems generally do not allow for proper drainage of urine and sufficient faecal matter disposal. Permanent or repeated contact of pigs with faeces increases the risk of infection between pigs in the same pen, especially enteric disorders [61]. The type of soil and building will also affect the ability to decontaminate the environment after the animals have passed through. Cleaning and disinfection followed by a fallowing period are the main elements of bio-compartmentalisation. Obviously, these techniques are difficult to apply in free-range farming [44] and remain complicated in litter farming where it is difficult to remove all organic matter. In addition, there are a limited number of disinfectant products that can be used in organic farming, especially in Europe [123].

### Summary (Table 2)

Biosecurity is probably the biggest challenge for alternative farming. In the last decades, major epizootics appeared in conventional farms. However, these farms have possibilities to control them, at least in the long term, by implementing strict internal and external biosecurity measures. In general, alternative farms apply biosecurity measures less strictly than conventional farms do. In free-range farms in particular, biosecurity measures are more difficult to implement, and fencing requires significant financial investment [96]. The development of alternative farms can represent a difficulty in the control of pathogens, in particular those circulating in wildlife. The difficulty of these farms in implementing strict biosecurity measures may be one of the obstacles to their development, as the risk to animal health may discourage farmers.

**Table 2.** Summary of the strengths and weaknesses for implementation of biosecurity measures according to production method.

| Abilities to implement | Conventional | Indoor with litter | Outdoor | Indoor with outdoor access | |
|---|---|---|---|---|---|
| General biosecurity | +++ | ++ | - | + | [100] |
| Bio-exclusion | | | | | |
| - Contacts: | | | | | |
| • Wildlife | +++ | ++ | - | + | [103,104,106–110] |
| • Pests and vectors | +++ | ++ | - | + | [97,109,111–114] |
| • Humans | ++ | ++ | - | + | [105] |
| - Litter and food | ++ | - | + | ++ | [109,115–117] |
| Bio-compartmentalisation | ++ | + | - | - | [44,61,100] |

+++ Very favourable; ++ fairly favourable; + not very favourable; - unfavourable

## 4. Animal Health in Alternative Farming Systems

Due in part to less easily implemented biosecurity and the biological diversity of the environment in which they live, animals raised in alternative systems are more likely to ingest pathogen-contaminated feed [124] and have more contact with pathogenic flora [125]. This vulnerability is accentuated in silvopastoral systems where there may be direct contact between pigs and wild boars or other wild animals that act as reservoirs for parasites or pathogenic microorganisms [124]. Thus, alternative systems are more likely to expose animals to infectious

agents. In addition, the husbandry systems and practices themselves can also be the cause of disorders, such as reproductive failures [126,127]. The strengths and weaknesses of alternative production systems on the health of animals are detailed in the following sections by major function, at all physiological stages.

### 4.1. Mortalities

Piglet mortality in the days following birth concerns all pig farms, but more particularly farms in alternative systems. Many studies have shown that perinatal mortality is higher in these types of production [59,128–131]. A comparison of three systems for farrowing Iberian sows [128] revealed that the survival rate of piglets born in a system in which sows are blocked during lactation is significantly better than those obtained either outdoors or in individual pens with straw and outdoor access for piglets. This mortality occurs mainly in the first four days of piglet life [129]. In organic production, early piglet mortality increases with litter size [60,83] and the age and the backfat thickness of the sow [60].

Piglet crushing is by far the leading cause of pre-weaning mortality [96,129]. One reason for higher crushing mortality in alternative systems may be that it is more difficult to monitor farrowing and intervene safely when a sow can move freely than when it is enclosed [132]. When farrowing occurs outdoors, much of the responsibility for monitoring farrowing and lactation is transferred from the farmer to the sow compared with sows in conventional systems [21]. However, this does not require the accrued presence of the farmer. In outdoor systems, farmer intervention during farrowing seems to have an unfavourable effect on pre-weaning mortality rates [59].

In a study of 112 farms in England [132], there were no differences in piglet mortality during the lactation phase regardless of whether the sows were in farrowing crates, free-ranged during all or part of the lactation or outdoors. This lack of difference can be attributed to various causes of mortality according to the type of farming: Crushing in alternative systems and due to infections indoors [132]. In the same study, there was a reduced risk of stillbirths in outdoor systems, which was attributed to the greater freedom of movement of sows resulting in faster farrowing. However, it was not possible to rule out a potential genetic effect.

For the subsequent post-weaning and fattening stages of rearing, mortality rates appear to be lower in alternative systems than in confined housing, although the post-weaning multisystemic wasting syndrome due to the porcine circovirus type 2 is expressed more severely in a straw-based housing system than in a fully-slatted housing system [61]. In a comparison of the influence of outdoor housing on animal health on 29 litters born indoors and 22 litters born outdoors, each litter was divided into two groups, one raised outdoors and the other indoors [133]. Pigs that were raised continuously or predominantly outdoors had a lower mortality rate.

The sow mortality rate appears to be higher in outdoor-reared sows [134], with less mastitis-metritis-agalactia syndrome (MMA) and intestinal torsion or distension, but more urogenital infections, heart failure and locomotor disorders [134].

### 4.2. Reproductive Disorders

Several studies suggest that sows kept outdoors have less MMA than those kept indoors [64,134]. One study found that sows kept indoors were more often treated for MMA problems than sows kept outdoors, and attributed that to the risk of constipation and soiling of the udder when the sows lack space [64]. However, it is probably more difficult for a breeder to observe MMA on a sow reared outdoors, which may explain why outdoor sows are treated less often than other sows.

Among the major pathogens to which pigs with outdoor access are more exposed than those without outdoor access is *Brucella suis*, which has a high seroprevalence in wild boars, particularly in Europe [107,135]. In a study carried out between 2000 and 2004 in Croatia, 67 of the 1997 herds tested positive to brucellosis by serology (3.4%). Of these, only two were not kept outdoors [135]. *B. suis* may cause reproductive disorders, such as fertility failures, abortions or orchitis in boars [136].

Leptospirosis also induces reproductive disorders in sows. Pigs with outdoor access are exposed to *Leptospira* serovars from wildlife [97], but there are few studies demonstrating overexposure of

these pigs. A study of 11 Swedish farms showed a relationship between rainfall and the level of seropositivity to *Leptospira interrogans* serovar (sv) Bratislava [137], suggesting that the risk of *Leptospira* infection for outdoor reared pigs increases with the amount of rainfall in a region. It is plausible that pigs kept indoors are not affected in the same way by variations in rainfall.

Reproductive performance is often lower in organic farms [41,126], but this does not necessarily mean that the health of the sows is poorer. Reproductive problems can also be related to the husbandry system. For example, it is known that sows kept in groups during lactation are more likely to ovulate during lactation [127] with reduced fertility after weaning.

### 4.3. Respiratory Disorders

Although open systems seem to offer many protective factors against respiratory diseases, particularly in terms of air quality, compared with confined conditions, respiratory disorders, although less frequent, are not totally absent in alternative systems [138]. Respiratory disorders are a problem mainly during the growing and finishing phases, rarely in breeding stock [38]. A study on sneeze and cough counts in 74 organic farms in eight European countries showed that there are fewer respiratory disorders when the pigs are raised with an outdoor run for their entire life [64]. However, it is more difficult to count coughs and sneezes in the open [38], which may lead to a bias in these results. To limit this bias, it is possible to compare lesions in the respiratory system at the time of slaughter. According to an expert panel convened by the European Food Safety Authority (EFSA), pigs kept outdoors generally have a lower prevalence of respiratory tract lesions during post-mortem inspections at the slaughterhouse than pigs kept indoors [61]. Several studies corroborate this observation and show a difference between conventional and alternative systems [139–142]. According to a study on 3054 pigs in Denmark [140], the real prevalence of respiratory lesions is 42% in conventional pig production compared to 16.5% in organic pig production, confirming results observed in a previous study on more than 200,000 pigs [141]. In Sweden, 7.4% of 3,963,799 conventional pigs observed at the slaughterhouse had pleuritis lesions compared to only 1.8% of 3464 organic pigs [142]. However, there are more recent discordant studies, e.g., in Denmark [40,41], that do not reveal significant differences in respiratory tract lesions between conventional, organic and free-range systems. Only pleuritis was more frequent in conventional farms in one of the two studies, with an estimated odds-ratio (OR) of 0.8 ($p < 0.0001$) for pleuritis lesions in alternative compared with conventional systems [51]. Beyond the production system itself, there may be a farming effect [50], in connection with different management of the risk factors for respiratory diseases, such as the quality of comfort and ventilation, hygiene, husbandry or even feeding [143–146].

### 4.4. Digestive Tract Disorders

Digestive troubles are another major disorder that predominates in pig production, as well as in alternative production systems—including organic production [21,38,64]. These diseases mainly affect suckling piglets and growing pigs, rarely sows [38]. However, only few studies have compared digestive disorders across different farming systems. One study found a higher frequency of diarrhoea in organic pigs when pigs were not kept outdoors [64]. In an outdoor system the access to a richer physical environment containing pasture, soil or other materials during lactation may increase feeding behaviour in the immediate post-weaning period, thereby reducing the period of starvation that usually accompanies weaning [84]. This starvation can have an impact on the gastrointestinal tract, such as villous atrophy [147], which contributes to the development of osmotic diarrhoea, while unabsorbed dietary material may act as a substrate for enterotoxigenic *Escherichia coli* in the gut. This may explain why less diarrhoea has been observed in this study. However, these results should be interpreted with caution, given the difficulty of observing this type of disorder in outdoor pigs [38,64]. Furthermore, increasing the age at weaning from 4 to 6 or 7 weeks, as in organic farms, does not seem to have an impact on functional characteristics of the small intestinal mucosa [89]. Beyond digestive disorders per se, no difference was found in the faecal flora in a study conducted on pigs at the end of the fattening period, regardless of whether they came from

conventional or organic farms [125]. Only the family *Bifidobacteriaceae* was significantly more frequent in the microbiota of organic pigs.

The observation of digestive lesions in the slaughterhouse shows that there are more enteritis or peritonitis lesions in organic production than in conventional production [140]. However, there are lower levels of parakeratosis or ulceration of the pars oesophagea in alternative systems (outdoor or straw) than in conventional farms [139]. However, in general, carcass inspection at the slaughterhouse is a low sensitivity technique for detecting digestive disorders [140].

As with respiratory pathology, beyond the production system itself, there is great variability in the expression of digestive disorders between farms, which can also be explained by differences in the management of risk factors, such as comfort, husbandry, feeding or hygiene. In the absence of certain raw (feed) materials, especially synthetic amino acids, feeding organic pigs can be difficult [148]. This requires particular attention in the choice of feed materials to prevent digestive disorders [148] and to ensure that animal welfare is not affected [149,150]. Hygiene plays a key role in the management of digestive disorders. According to Vannucci et al. [151], necrotic enteritis induced by *Lawsonia intracellularis* is more frequent under conditions that facilitate faecal–oral cycling, such as use of straw bedding or solid flooring with poor sanitation.

*4.5. Parasitism*

The presence of internal parasites and the intensity of infestations are strongly influenced by the production system [152]. Although a decrease in the number of parasite species and infestation pressure has been observed in conventional farms [152], internal parasitism remains a major preoccupation in alternative systems, especially production systems with outdoor access [16,131,138]. These rearing conditions are indeed more favourable for the development and survival of the different stages of parasites in the environment [124].

In a study carried out in the Netherlands on 9 conventional, 11 organic and 16 non-organic farms with outdoor access [153], the prevalence of helminth infestations was higher when pigs were kept outdoors, independently of the organic status of the farm. However, parasite diversity remained low. In that study, only three helminths (*Ascaris suum*, *Oesophagotomum* spp. and *Trichuris suis*) and coccidia were identified. This result has been echoed in multiple studies [153–155] in which mainly these parasite species are isolated, with varying prevalence rates. Other species such as *Hyostrongylus rubidus*, *Metastrongylus* spp., *Strongyloides ransomi* or *Stephanurus dendatus* are less frequently observed [155–157].

Coccidiosis can be an important cause of diarrhoea in suckling piglets over 7 days of age. It is usually caused by *Cystoisospora suis*. In the above-mentioned Netherlands study, there were systematically more oocysts present in the faeces of sows in the alternative systems than in the conventional systems [153]. However, these were *Eimeria* spp. oocysts whose impact on pigs remains to be assessed. *C. suis* was isolated in piglet faeces, with no significant differences according to production system.

Helminths constitute a pivotal issue in alternative farms. In the Netherlands study, there was no difference in the levels of *Oesophagotomum* spp. infestation between farming systems [153]. Approximately 25% of farms had *Oesophagotomum* spp., with infestations becoming more frequent with age, confirming observations from other studies [152,155,157]. Regarding *T. suis*, the level of infestation was significantly higher (37%) in free-range farms than in conventional farms (11%) [153]. The predominantly infected stages of rearing vary between studies. In the Netherlands study, the level of carry-over was higher in free-range sows (39%) [153], whereas it was higher in post-weaning piglets in a study of 20 organic farms in France [157]. In case of massive infestation, *T. suis* may be responsible for weight loss or even haemorrhagic diarrhoea [16,44]. *T. suis* is the second most frequently found endoparasite, far behind *A. suum* isolated in 60% of alternative farms, compared with only 11% of conventional farms [153]. Piglets in post-weaning and fattening pigs are mainly affected [153,157]. *A. suum* migrates through the liver during larval migration, resulting in the formation of "milk spots" which are easily observed in the slaughterhouse if the migration occurred in the month prior to slaughter [158]. Many studies have compared the prevalence levels of livers

with "milk spots" according to farming system. In almost all studies, the frequency of affected livers is higher when pigs come from alternative farming systems with outdoor access [50,51,140,159]. In particular, in a Danish slaughterhouse study, pigs were much more likely to have milk spots if they were kept outdoors, in organic or conventional systems: 15% of the pigs with outdoor access had liver lesions compared with only 4.6% of the pigs kept indoors [50]. However, a significantly different observation has been reported from a slaughterhouse study, in which pigs from non-organic farms had more liver lesions (5.6% versus 4.1% of organic pigs) [142]. In the same vein, Sanchez-Vazquez et al. [160] showed that the presence of bedding on a solid floor (e.g., straw) was a risk factor (OR = 1.5 [1.26; 1.85]$_{95\%}$) for a high prevalence of milk spots, although farms with all stages of production indoors had a lower risk (OR=0.4 [0.32; 0.49]$_{95\%}$).

Parasitosis control is based on breaking the life cycle of the parasite, especially in the environment. Improved hygiene and the use of slatted floors have reduced faecal–oral contamination and parasitism problems in confined farms [152,159]. Breaking the cycle is complicated in alternative farms due to the nature of the environment. One study found *A. suum* eggs in 14 and 35% of pastures hosting organic sows and pigs, respectively [155]. More recently, a study on organic farms in Sweden [161] found *A. suum* and *T. suis* eggs in respectively 79% and 57% of 28 soil samples. Farms that have been keeping pigs outdoors for a long time appear to be more infested by this parasite [157,161], probably due to the resistance of *A. suum* eggs, which can remain viable for up to 10 years in the environment [156]. Piglets born outdoors may thus be highly infested earlier in their lives than piglets reared indoors without any outdoor access [159]. The control of internal parasitism is essential in alternative production to maintain animal health and welfare. Pasture rotation and stocking rate are important factors influencing parasite pressure and infestation risk [124]. Rotation programmes should include all production aspects, including housing [162]. However, although hygiene, rotational pasture and low stocking rates are appropriate measures to control parasite pressure, the use of anthelmintics remains the most important measure to control parasitosis [156], because the application of non-medical prophylaxis only may have a limited effectiveness. Nonetheless, the use of chemical control poses other problems with regard to the environment and the development of resistance.

The main ectoparasites in pigs are scabies (*Sarcoptes scabiei* var. *suis*) and lice (*Haematopinus suis*), although fleas or ticks can also be detected, especially in pigs with access to extensive pastures or woods [124]. Depending on the study, various prevalence levels have been reported. In 29 of 48 Austrian organic farms studied, *H. suis* and *S. scabiei* (60%) were found [154], but a study on nine Danish organic farms found no signs of scabies or lice [155]. A survey conducted in 110 German farms reported that 2.5% and 19.1% of the observed sows were infested with *H. suis* and *S. scabiei* var. *suis*, respectively, with a higher *S. scabiei* infestation risk for sows housed on straw in farrowing pens (OR = 15 [2.9; 77.6]$_{95\%}$) [163]. For *H. suis*, the access of sows to the outdoors was an important risk factor (OR = 12.7 [4.0; 40.7]$_{95\%}$).

### 4.6. Skin Disorders

Apart from lesions induced by fights (scratches, wounds), or parasitosis, skin disorders in organic pigs seem to be rare [41]. A Danish slaughterhouse study reported that pigs with outdoor access had three times more skin lesions than conventional [50]. Nevertheless, there was no differentiation in the type of lesions, consisting mainly of wounds, dermatitis, eczema and insect bites. It seems likely, however, that some lesions, such as sunburn, are more frequent in pigs kept outdoors [50]. An earlier study also observed more eczema and insect bites in animals from unconventional farms (OR = 3.2, $p < 0.001$) [51].

### 4.7. Locomotor Disorders

Locomotor disorders, primarily due to the presence of lameness in pigs, are often observed on farms, particularly in organic production [113]. They affect all stages, including breeding stock.

Several studies show that sows raised in alternative systems have a lower risk of lameness. In one study on 1054 pregnant sows from 44 conventional and 9 organic farms, 24.4% of the

conventional sows had lameness, but only 5.4% of the organic sows were lame, although with a seasonal effect (more lameness in summer and autumn) [164]. Soil quality seems to be important. In a study involving 646 sows from 21 herds [165], sows housed on slatted floors had twice the odds of being lame and 3.7 times the odds of being severely lame than sows housed on solid floors. Another study observed less lameness when sows had outdoor access, linked, according to the study's authors, to softer bedding, less exposure to manure and increased activity [64].

Lameness in animals can have different causes, such as arthritis, fractures, foot and claw lesions. Osteochondrosis is considered to be the main cause of leg weakness in pigs [44,166] and consequently lameness. Osteochondrosis is a lesion traditionally observed between 6 and 20 weeks of age, but can also affect older pigs, especially young breeding stock [44,167]. It is a widespread lesion on farms [44]. Another study corroborated the pervasiveness of osteochondrosis, with 41.4% of 345 pigs observed at the slaughterhouse having osteochondrosis lesions, 12.4% had severe lesions; pigs from conventional farms had significantly more lesions than those raised on litter [166]. However, this difference has not been observed in all studies. For example, one study found no differences in the prevalence of osteochondrosis between animals raised on slatted, solid or straw floors [168]. In contrast, due to the higher number of condemned joints in organic free-range pigs than in conventional pigs, a Swedish study set out to compare the joints at the time of slaughter of 91 pigs reared on straw and outdoor runs with those of 45 pigs reared confined on partial slatted floors [169,170]. The study found that there were more frequent and more severe osteochondrosis lesions in livestock raised with outdoor access. This result was attributed to the development of osteochondrosis being favoured by the magnitude and diversity of biomechanical stresses in outdoor pigs.

Arthritis is another cause of lameness. Observations at the slaughterhouse show that arthritis is more common when animals are raised outdoors (organic or not) [50,51]. One explanation is the difficulty in treating sick animals in these production systems. In an abattoir study of nearly 4 million pigs, there was more arthritis in organic than in non-organic pigs (1.5% versus 0.4% in conventional pigs) [142]. Poorer management of *Erysipelothrix rhusiopathiae* in organic pigs, a bacterium that can reside in the soil of the pens, and a lower vaccination rate are among the explanations. There are probably other intrinsic factors in outdoor farming that have not been identified.

Foot injuries, in which the quality of the floor plays a major role, are another cause of lameness [61]. One study showed that lameness in sows is positively correlated with heel and claw lesions [171]. Concrete slatted floors are a significant risk factor for lameness compared to straw floors (OR = 9.9 [4.4; 34.5]$_{95\%}$). Another study observed the claws of 245 pigs raised on straw, solid floor or slatted floor in the slaughterhouse, finding more lesions on the volar surface or at the white line area of the claws of pigs that had been raised on a solid floor without straw, owing to an accumulation of urine and manure and a more slippery floor [168]. That study also had a density effect, with significantly more lesions in general, and more particularly at the white line, when 0.65 m² per individual was provided instead of 1.2 m².

### 4.8. Anaemia and Iron Supplementation

Piglets are naturally born with few iron reserves. Anaemic piglets, which are iron-deficient, are more susceptible to infection, because iron plays a role in the pig immune system [172,173]. To compensate for this deficiency, iron is usually administered intramuscularly or orally in conventional pig farming. However, iron supplementation is rarely administered in outdoor or organic production [21]. It is often agreed that piglets born outdoors ingest sufficient iron from the soil and do not need supplementation [174]. However, several studies have shown that iron supplementation may still be useful or even necessary in free-range-born pigs. One study showed higher weaning haemoglobinemia in piglets born outdoors and given 200 mg intramuscular iron at 2–4 days of age, than in unsupplemented piglets, with no effect on animal performance [71]. Another study showed that supplemented piglets born outdoors had higher haemoglobinemia and body weight at weaning and lower pre-weaning morbidity and mortality than unsupplemented pigs [175]. Finally, a more recent study showed that a second iron intake at about 2 weeks of age can even be useful in organic

piglets weaned at about 42 (± 1.7) days of age, to ensure adequate iron intake and better piglet growth until weaning [176]. It is therefore probably appropriate to assume that the risk of anaemia is low in free-range farms where the soil provides iron in its intrinsic composition [41].

### 4.9. Immunity

Differences in immune responses appear to exist between livestock reared outdoors and those reared indoors. Indeed, one study showed that pigs kept outdoors had lower white blood cell counts, lymphocyte levels and natural killer cell responses, and higher neutrophil levels than pigs without outdoor access [174]. Another study showed differences in immune responses to bovine thyroglobulin injections between pigs kept indoors and those with outdoor access [177]. Immunoglobulins G and M increase less rapidly following injection for pigs with outdoor access due, according to the study's authors, to differences in infection pressure, environmental differences, including temperature, and stress levels.

Exposure to certain toxic substances can affect the immune defence capabilities of animals. Some mycotoxins, such as deoxynivalenol or fumonisins for example, have a negative impact on the immune system of pigs, with increased susceptibility to infectious diseases and decreased vaccine efficacy [178,179]. Mycotoxins are metabolites produced by various species of fungi that grow on many raw materials used for animal feed. Because organic livestock production cannot use fungicides, higher levels of mycotoxins may be present in cereals [180], as demonstrated in one study, with higher levels of deoxynivalenol contamination in organic wheat than in conventional wheat [36]. In addition, the straw provides also a risk of exposure to mycotoxins, especially those produced by *Fusarium* [61].

Immune system functions require energy, protein, vitamins and trace minerals [181]. The ban on the use of synthetic raw materials (amino acids, vitamins, etc.) in organic farming requires great care in the choice of raw ingredients to avoid negative consequences on the immune system of animals.

### 4.10. Treatment and Prevention

The integrity and effectiveness of the immune system is particularly important in organic production because treatment options for infections are limited [123,182]. In the European Commission regulation regarding organic production, it is stipulated that if a health problem arises, animals should be treated, in order of preference, with homeopathic dilutions of substances of plant, animal or mineral origin, plants or plant extracts without anaesthetic effects, or substances such as trace elements, metals, natural immunostimulants or authorised probiotics [123]. The use of synthetic allopathic treatments, including antibiotics and anthelmintics, is permitted but only to a limited extent [123,182]. However, scientific evidence of the effectiveness of these alternative treatments, particularly homeopathy, is often missing or incomplete [87,183]. Using an insufficiently effective treatment on a sick animal obviously affects its health and well-being [149]. Moreover, organic farmers are less likely to call a veterinarian to treat health problems than their conventional colleagues [87]. Nevertheless, the results of a survey carried out between 2007 and 2010 on 104 European organic farms (Corepig programme) indicate that the use of antibiotics and anthelmintics may not be so rare on these farms. A wide variety of practices exist between countries and between farms [70]. Although the use of antibiotics in organic pigs is obviously lower than in conventional pig herds [131], antibiotic treatments have been administered to post-weaning pigs in more than half of the farms surveyed [70]. Deworming of post-weaning pigs, and mainly of sows, is very common in Germany, France and Austria. The use of allopathic chemical treatments in organic farms, when necessary, may partly explain the lack of differences in the prevalence of lesions observed at the slaughterhouse in several studies [50,51] between organic and non-organic outdoor farms. The difficulty in treating free-range animals may also explain, at least in part, the higher prevalence of many lesions observed in these studies in both organic and non-organic free-range systems compared with conventional systems.

Summary (Table 3)

As with animal welfare, alternative production systems have advantages and disadvantages in terms of animal health: Outdoor access overexposes animals to diseases carried by wildlife, especially wild boar (e.g., brucellosis) and to pathogens that are almost impossible to control under these conditions. Unlike the conventional rearing system where it is possible for most of the pathogens to sanitise the environment, it is much more complicated in systems on straw and almost impossible outdoor. This explains, at least in part, why parasitism is so difficult to manage in these farming systems. The practical difficulty of treating sick animals outdoors is another problem, which contributes, in particular, to the increase in infection pressure in paddocks. However, the lower density of animals in this type of production counterbalances this potential increased infection pressure. Pigs kept in these farming systems appear to be less susceptible to respiratory diseases, linked to a lower exposure to risk factors associated with confinement. The comfort provided to the animals, especially to the breeders, contributes to limit leg disorders, but the management of piglet crushing in farrowing units is a challenge. Those weaknesses are not unavoidable, with appropriate deworming plans and good biosecurity measures set up, these farming systems have the opportunity to rear their animals in good health condition.

**Table 3.** Summary of the strengths and weaknesses in terms of animal health according to production method.

| | Conventional | Indoor with litter | Outdoor | Indoor with outdoor access | |
|---|---|---|---|---|---|
| Mortalities: | | | | | |
| - Sows | +++ | ? | - | ? | [134] |
| - Suckling piglets | | | | | |
| • Stillbirths | - | ++++ | +++ | +++ | [132] |
| • Crushing | +++ | - | - | - | [128,132] |
| • Infectious causes | - | +++ | +++ | +++ | [132] |
| • Predation | +++ | +++ | - | - | [95,96] |
| - Growing Pigs | - | ? | +++ | +++ | [133] |
| Reproductive disorders: | | | | | |
| - Brucellosis | +++ | +++ | - | - | [135] |
| - Mastitis-Metritis-Agalactia Syndrome | - | ? | ++ | ++ | [64,134] |
| Respiratory disorders | - (+/-) | ? | +++ (+/-) | +++ (+/-) | [61,64,139–142] ([50,51]) |
| Digestive disorders | | | | | |
| - In general | +/- | +/- | + | + | [84,89,125,140,151] |
| - Stomach ulceration | - | +++ | +++ | +++ | [139] |
| Locomotor disorders | | | | | |
| - Sows | - | +++ | +++ | +++ | [64,164,165,171] |
| - Osteochondrosis | +/- | +/- | +/- | +/- | [166,168–170] |
| - Arthritis | +++ | ? | - | - | [50,51,142] |
| Skin disorders | +++ | ? | - | - | [50,51] |
| Parasitism | | | | | |
| - Internal | +++ | + | - | - | [50,51,140,153,155–157,159] |
| - External | +++ | - | - | - | [163] |

+++: Very favourable; ++ fairly favourable; + not very favourable; - unfavourable; +/- divergent opinions; ? unknown.

## 5. Pork Safety in Alternative Farming Systems

Foodborne zoonoses are infectious diseases of major health and economic importance in developed countries [184]. Pigs represent a reservoir of many bacterial, viral and parasitic pathogens [124]. The presence of zoonotic pathogens, their reservoirs or their vectors in or near the direct environment of pigs can lead to high prevalence levels of these infectious agents in alternative livestock populations.

*5.1. Bacterial Contaminations*

The four most frequent and/or serious bacterial hazards are *Campylobacter* spp., *Salmonella* spp., *Yersinia enterocolitica* and *Listeria monocytogenes* [184]. Campylobacteriosis was the most frequently reported zoonosis in Europe in 2018 [185]. It is caused by a bacterium that is widely distributed in pig faeces [186] which is not, however, the main source of contamination for humans [185]. Human infections are mainly caused by *Campylobacter jejuni* [185] while *Campylobacter coli* is the most common species in pigs [187]. According to one study, there is no significant difference in the prevalence of *C. coli* carriage between pigs from organic and non-organic farms [188]. In contrast, with regard to *L. monocytogenes*, a Finnish study showed that the prevalence of the bacterium was significantly higher in organic pigs than in conventional pigs, attributing it to *L. monocytogenes* being a common bacterium in the environment, making outdoor areas a potential source of contamination [189]. However, in that study, the difference between organic and conventional pigs disappeared when the analysis focused on the contamination of carcasses, because high prevalence of contaminated pigs in a farm does not necessarily lead to highly contaminated meat [189]. Contamination of meat with *L. monocytogenes* is mainly a slaughterhouse issue [190]. The levels of *L. monocytogenes* contamination of pork, which is often reported in the media, are however generally low [185] and cause few foodborne zoonoses, unlike *Y. enterocolitica*. During the 2005–2018 period, pork meat and products thereof were among the most frequently reported food categories causing foodborne yersiniosis [185]. *Y. enterocolitica* is a widely distributed bacterium on farms [191,192]. According to a study in the Czech Republic, there seems to be no difference in seroprevalence between conventional and organic farms, with small family farms having significantly lower seroprevalence [191].

However, several studies have shown that the prevalence of *Y. enterocolitica* in conventional farms is higher than in alternative farms. In particular, in a German study [193] including 210 pigs from six conventional fattening farms and 200 pigs from three alternative farms, the prevalence of PCR positive animals for *Y. enterocolitica* was between 20% and 37% in conventional farms, and between 16% and 23% in alternative farms. The prevalences in samples taken from animals from alternative housing were significantly lower than those from conventional housing. In a more recent study [194] on 788 pigs from 120 Finnish farms, the authors showed that the organic production type was one of the most significant protective factors for *Y. enterocolitica* contamination, linked, according to the authors, to the generous use of bedding, limited use of antibiotics and lower animal density. This study is in line with the results of another Finnish study [195] where the prevalence of *Y. enterocolitica* was significantly higher in all sample types from conventional pig production than in organic pig production. In this study the high prevalence of *Y. enterocolitica* on farms was also associated with the absence of bedding. In contrast, a study of 287 Norwegian farms showed that the use of straw as bedding for fattening pigs was a risk factor for *Y. enterocolitica* seropositivity (OR = 2.25 [1.04; 4.89]$_{95\%}$) [192]. In another study conducted in Germany on 80 fattening pig farms [196], housing on a fully slatted floor was observed more frequently in herds with low serological *Yersinia* prevalence. Based on these studies, it therefore appears that the prevalence of *Y. enterocolitica* is lower in organic farms but that straw may be a risk factor, which may seem contradictory. However, it should be noted that studies highlighting the role of straw have only been carried out in conventional farms.

The subject of *Salmonella* infections in pigs is well documented, with extensive research being carried out for many years studies on *Salmonella* spp., although most of the time there is no clinical expression of the disease in animals [61]. Salmonellosis is the second most common cause of bacterial foodborne outbreaks in humans [112,185]. Although egg products are the main source of infection, pork was responsible for 5.4% of foodborne *Salmonella* spp. outbreaks in 2018 in Europe [185]. Many measures at the farm level can reduce the prevalence of *Salmonella* spp. and therefore the risk of cross-contamination of carcasses at the slaughterhouse [112]. However, given the complexity of salmonella epidemiology [197], it is not always easy to establish the role of the farming system in controlling the bacteria and the results are sometimes contradictory. For example, a Danish study found no differences in the seropositivity levels obtained from meat juice samples from organic (11 farms), conventional (11 farms), or non-organic farms with outdoor access for pigs (12 farms) [198]. However,

most studies tend to show that alternative organic and/or outdoor systems are a risk factor for *Salmonella* spp. Another study in Denmark showed that pigs from farms with free-range access to at least one category of pigs in the farm had a higher risk of being seropositive for *Salmonella* spp. (meat-juice samples), than pigs from conventional farms [199]. However, in that study, if a category of pigs had access to a free-range area, the farm was classified as free-range; therefore, there were farms where only sows were kept outdoors. In a study conducted in the Netherlands, the seroprevalence in blood samples collected at the slaughterhouse was significantly higher if non-organic finishing pigs had free-range access (44.6% compared to 24.5% for conventional pigs) [200]. The study's authors found no significant difference between organic farms (37.5%) and the other two categories, possibly due to the low number of samples collected from organic farms [200]. Based on 675 sera collected in three American states (Wisconsin, North Carolina and Ohio), there was significantly higher seroprevalence of *Salmonella* spp. in "antimicrobial-free" farms, i.e., with pigs having outdoor access and not receiving antibiotic growth promoters [201]. Similarly, in faeces collected from 44 farms, *Salmonella* spp. were significantly more frequently isolated from organic farms (79% of farms) than from conventional farms (53% of farms) [202]. After PCR serotyping, there was a greater diversity of serovars isolated from organic farms, even though the *Salmonella* sv Typhimurium was significantly more frequently isolated from conventional farms.

Floor type appears to be of great importance, because the permanent and repeated contact of pigs with faeces increases the risk of faecal–oral contamination and salmonella infections [61]. This contamination was demonstrated in a study in which mesenteric lymph nodes of pigs from 62 farrow-to-finish pig herds were sampled and analysed bacteriologically: 57 farms (91% of the farms) had at least one positive sample [203]. Floor type emerged as a risk factor, with all farms in which slatted floors represented less than 50% of the floor area having positive samples. In the Netherlands study, the authors explained that the presence of full, unslatted floors with straw in unconventional farms may partly explain the differences in observed seroprevalence [200]. An original study showed that *Salmonella* can persist in the environment, especially in outdoor areas. *Salmonella* could be isolated from soil samples up to 5 weeks after the departure of contaminated pigs, and during the seven-week study period in shelter huts [204]. The introduction of disease-free pigs into these *Salmonella*-contaminated paddocks led to the contamination of some of them. Outdoor wallows may also be a source of contamination for outdoor pigs and a source of persistence of the bacteria in the environment [205]. Finally, in a study in which bird droppings, pig faeces and environmental samples (soils, water puddles and farm equipment) were analysed, birds also clearly contribute to the persistence of the bacterium in outdoor areas [206]. Monophasic *Salmonella* Typhimurium DT193 was the most commonly isolated serotype in all three types of samples. The study's authors explained that this serotype is generally associated with pigs, suggesting that pigs were the initial source of infection. Interestingly, environmental samples were positive, even in a field that had not been occupied by pigs for more than two years.

*5.2. Parasite Contaminations*

Changes in methods of production to closed systems have nearly eliminated the risk of *Taenia solium*, *Trichinella spiralis* and *Toxoplasma gondii* in pork from conventionally raised pigs [197]. However, due to higher exposure and less strictly observed prevention measures in alternative farming systems, the risk of foodborne zoonoses of parasitic origin may increase again.

Improved animal husbandry practices, meat inspection, consumer education and medical care have, in particular, significantly reduced the incidence and health impact of trichinellosis in humans [207]. However, pork remains the cause of many outbreaks, mainly in Eastern Europe and Argentina, where traditional "backyard" pig farming is still very popular [207]. Many mammal species are susceptible to *T. spiralis*, such as foxes, rats, raccoons and wild boars. Pigs are infested mainly by mouth, by ingestion of larvae from muscles of parasitised animals. It therefore seems inevitable that pigs with outdoor access are at greater risk of infection by *T. spiralis* by being more frequently and intensely in contact with reservoirs. However, this risk appears to be low given the prevalence or seroprevalence levels estimated in many studies. For example, one study in the United States found

only 2 seropositive samples out of 675 samples (0.3% of samples) [201]. In a study in the Netherlands, only 1 to 3 pigs out of 845 were seropositive (0.12% to 0.35% of samples, depending on the seropositivity cut-off) [208]. However, in both studies, seropositive individuals were always from organic or equivalently certified organic farms with outdoor access, never from conventional farms. These results are corroborated by those from a study in Argentina on 21 pig farms from which muscles and blood samples from 3224 pigs were collected and tested for the presence of *T. spiralis* larvae or antibodies: 67 pigs (2.1%) showed positive samples, all from six farms rearing pigs outdoors (28.6% of the farms) [209]. None of the pigs sampled from the seven confined or semi-confined farms were infested with the parasite in that study and the prevalence of *T. spiralis* was significantly higher on farms where pigs had access to wildlife carcasses.

*Toxoplasma gondii* is a widespread protozoan that affects both animals and humans. One of the main routes of human infection is the consumption of raw or undercooked meat from certain animal species, including pigs. Pigs can become infected with *T. gondii* through the ingestion of food or water contaminated with sporulated oocysts, through consumption of cysts contained in the tissues of infected animals such as rodents, birds and other pigs, or congenitally [210]. As with *T. spiralis*, numerous studies around the world show that the prevalence of *T. gondii* in pigs in confinement is low [201,208,211], and that access to an outdoor area is a major risk factor for pigs [183,201,208,210–212]. In a study conducted in Sweden on the sera of 972 pigs, including 326 conventional and 646 alternatively raised pigs, there was a significant difference in *T. gondii* seroprevalence between conventional pigs (1.2% [0.3; 3.1]$_{95\%}$) and pigs raised in alternative systems (7.9% [5.9; 10.2]$_{95\%}$) [211], confirming the observations from studies in the United States [201] or in the Netherlands [208]. They also estimated that the risk of being seropositive increased with the length of time of access to an outdoor area (OR = 1.8 [1.3; 2.7]$_{95\%}$ for a one-month increase in the length of pasture exposure). In the Netherlands the risk of detection of *T. gondii* antibodies in a free-range farm was almost 16 times higher (OR = 15.8 [2.0; 124]$_{95\%}$) than in an indoor farm [208], a finding more recently confirmed in a study in France, which estimated that pigs kept outdoors are 3.6 times more likely to be seropositive (OR = 3.62 [1.94; 6.59]$_{95\%}$) [212]. Access to outdoor installations may facilitate contact with cats and/or rodents, increasing the chance of ingestion of oocysts and tissue cysts by pigs [210]. A study in the USA estimated that sows in contact with cats were nearly three times more likely to be seropositive than those not exposed to cats (OR = 2.6 [2.0; 3.4]$_{95\%}$) [213]. Iberian sows [214] also showed an association between *T. gondii* seroprevalence and the presence of cats. In these two studies, access to an outdoor area is also a risk factor for sows, as it is for fattening pigs, the American study estimated that sows with an outdoor access were 23 times more likely to be seropositive than confined sows (OR = 23 [16.5; 35.4]$_{95\%}$) [213]. In general, seropositivity levels are higher in sows [212,215], with seroprevalence levels as high as more than 30% in some studies [213,215].

## 5.3. Hepatitis E Virus

The frequency of sporadic cases of hepatitis E in humans has increased in the last few years in developed countries. The consumption of raw or undercooked pig liver-based products has been identified as an important source of human infections [216]. Seroprevalence levels in pig farms can be very high [108,216,217]. A recent study showed that seroprevalence in Corsica (France) reaches 85.4% [79.8%; 91.0%]$_{95\%}$ in domestic pigs, which are mostly raised under extensive conditions [108]. In addition, they showed that pigs kept in free-range or fenced pens had much higher seroprevalence levels than pigs kept under intensive confinement conditions (OR = 10.1 [2.6; 38.8]$_{95\%}$). The study's authors concluded that extensive farming practices may play a role in virus exposure. Another study comparing seroprevalence obtained in conventional, organic and non-organic farms with an outdoor area found that seroprevalence was significantly higher (89%) only in organic farms than in conventional farms (76%) [217]. This difference may be due to housing conditions that allow greater exposure of animals to manure, thus increasing the possibility of virus transmission [217]. This hypothesis should be considered in conjunction with the results of the study by Fernández-Barredo et al. [218],who isolated virus RNA from raw manure from 8 of the 16 farms surveyed.

### 5.4. Antibiotics and Antibiotic Resistance

Consumers often perceive meat produced in alternative systems as more nutritious and safer, with less use of feed additives and antibiotics [30]. Consumption of synthetic chemical drugs is indeed lower in external production systems [30]. In organic livestock farming in particular, the use of medicines and antibiotics is limited and is one of the requirements for organic certification. For example, antibiotic use in Denmark in 2016 was 10 times lower in weaners and fatteners on organic farms than on conventional farms [219]. Moreover, according to the European rules for organic production, the withdrawal period between the last administration of a chemically synthesised allopathic medicinal product and the slaughter of the treated animal is doubled compared to the legal withdrawal period. If this period is not specified, it is set to 48 h [123]. The risk of persistence of residues of these treatments in products derived from these animals is therefore extremely low. In a study conducted in the Netherlands, no antibiotic residues were detected in kidney and meat samples collected at the slaughterhouse from organic pigs [186].

Knowledge is still limited on the transmission of antibiotic-resistant bacteria or genes to humans via meat [220]. However, transmission through food must certainly play a role in the global epidemiology of antibiotic resistance [197]. Numerous studies have focused on antibiotic resistance in bacteria found in animal husbandry, with the aim of comparing them between types of farming, mainly between conventional and organic farms. Almost all studies show that the prevalence of antibiotic-resistant bacteria is lower in organic farms, probably due to the lower use of synthetic antimicrobials. For example, in a study of 223 farms (111 organic and 112 conventional) in four European countries (Denmark, France, Italy and Sweden), differences in antibiotic resistance were found between *Escherichia coli* isolated from colon contents or faeces of conventional pigs and those isolated from organic pigs [221]. The percentages of strains resistant to ampicillin, streptomycin, sulphonamides or trimethoprim were significantly lower for *E. coli* from organic pigs. Other studies confirm these observations [186,222] for *E. coli* but also for other bacteria such as *Salmonella*. A study conducted in Korea between 2012 and 2013 showed that *Salmonella* isolated from organic farms were significantly less resistant to tetracyclines, ampicillin or gentamycin [202]. In addition, in that study there were significantly more salmonella resistant to three or more classes of antimicrobials in conventional farms. Another study conducted both in France and Sweden found differences in resistance to tetracyclines and erythromycin among *Campylobacter coli* strains isolated from colon contents or faeces, again in favour of pigs from organic farms [188]. However, these results differed between countries, with differences being significant in France, but not in Sweden. These differences between countries have also been reported elsewhere [221]. For example, the proportion of tetracycline-resistant *E. coli* was significantly lower in organic pigs than in conventional pigs in France, Italy and Denmark, but not in Sweden. For some antibiotics, the proportion of resistant *E. coli* from conventional pigs in one country was even lower than for organic pigs in another country, which, according to the study's authors, is related to lower antibiotic exposure in conventional farms in some countries, such as Sweden. This may also explain why one study did not find significant differences between farming systems in a study to determine the abundance of antibiotic resistance genes (*sul1*, *sul2*, *strA*, *tet(A)*, *tet(B)* and *cat*) within faecal microbiota in pigs kept under conventional and organic farming systems in Denmark, France, Italy and Sweden [125]. The only significant difference was in the abundance of antibiotic resistance genes in the samples from different countries, with a higher abundance in southern European countries (France and Italy). In that study, geographical location had more influence on the antibiotic resistance of the faecal microbiota than farm's status as conventional or organic.

Summary (Tables 4 and 5)

Although the lower use of chemical allopathic treatments, and in particular antibiotics, helps to limit the risk of human contamination by residues or antibiotic resistance through the consumption of pork from alternative pig farms, the higher prevalence of many zoonotic pathogens in this kind of farms represents a risk both for the consumer and for the image of the alternative systems which

today enjoy a very favourable perception on these issues. Except for *Yersinia enterocolitica*, which appears to be less prevalent in organic farms, access to an outdoor area is a risk factor for pigs being infected by foodborne zoonotic pathogens, of bacterial, parasitic or viral nature. The difficulty of sanitising the environment, especially the soil outdoors, increases the risk of contamination of free-range pigs by these pathogens. In addition, pigs reared outdoors are more easily in contact with vectors or reservoirs, such as birds, rats, foxes or wild boars. It is therefore important for the alternative production systems to set up the necessary hygienic and biosecurity measures to limit pig contamination as far as possible, but also to comply with well suited control plans in order to minimise the risks of human contamination through the consumption of pork.

**Table 4.** Summary of the strengths and weaknesses in terms of health safety according to production method.

| | Conventional | Indoor with litter | Outdoor | Indoor with outdoor access | |
|---|---|---|---|---|---|
| *Campylobacter coli* | ++ | ++ | ++ | ++ | [188] |
| *Listeria monocytogenes* | +++ | ? | - | - | [189] |
| *Yersinia enterocolitica* | ++ | + | +++ | +++ | [191–196] |
| *Salmonella* | +++ | + | - | - | [199–201,203–206] |
| *Trichinella spiralis* | +++ | ? | - | - | [201,208,209] |
| *Toxoplasma gondii* | +++ | ? | - | - | [183,201,208,210–212] |
| Hepatitis E virus | +++ | ? | - | - | [183,201,208,210–212] |

+++: Very favourable; ++ fairly favourable; + not very favourable; - unfavourable; ? unknown.

**Table 5.** Summary of the strengths and weaknesses in terms of antibiotic use and antibiotic resistance according to production method.

| | Conventional | Organic | |
|---|---|---|---|
| Antibiotic and antibiotic resistance | - | +++ | [186,188,202,221,222] |

+++: Very favourable; - unfavourable.

## 6. Conclusions and Outlook: Main Challenges for Alternative Systems

It is difficult to define a single alternative farming system, because they are so diversified, ranging from farming on straw to silvopastoral farming, organic farming or free-range farming, but they all differ to conventional, slatted, confined farming and enjoy a very positive societal image. These farms have real strengths, but they also have weaknesses, constituting major challenges to be met. Controlling biosecurity is undoubtedly the most important, and one of the most difficult challenges, but necessary to prevent contamination of livestock farms, which has an impact on animal health but also on the safety of the meat produced there. Paradoxically, efforts must also be made to improve animal welfare, even though the consumer is in favour of these systems for this reason in particular. All these challenges need to be considered as a in an integrated, holistic manner, following good practices and the know-how of the farmer being central to the success of these farms. Finally, these alternative farms are above all enterprises, in which the welfare of the farmer and the economic profitability are also priorities, as well as environmental concerns. To ensure the sustainability of these farms, it appears that a more global assessment of all these criteria is necessary and that the sole consideration of the highly mediatised societal concerns related to animal welfare and environmental issues only leads to a partial and biased evaluation of these farming systems.

**Author Contributions:** M.D. reviewed literature and drafted the manuscript. C.F. and F.P. supervised the project. All co-authors revised the manuscript. All authors have read and agreed to the published version of the manuscript.

**Funding:** This research received no external funding

**Conflicts of Interest:** The authors declare no conflict of interest.

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
