# Peer review of "Pig Farming in Alternative Systems: Strengths and Challenges in Terms of Animal Welfare, Biosecurity, Animal Health and Pork Safety"

_agriculture, doi:10.3390/agriculture10070261_

Round 1

Reviewer 1 Report

The title of the manuscript is misleading, as there are very few "opportunities" explained in the manuscript. Most of the text is a plain listing of mostly weaknesses in alternative systems. There are too many keywords listed(11), and throughout the manuscript several different fonts are used for citation numbers. Abstract is well organized and appropriate.  

The overall idea of the manuscript is interesting and review of this type is always a good addition to the field. Unfortunately, this manuscript lacks several important points and that is the reason why it needs major revision:

The WHOLE MANUSCRIPT - Generally throughout the manuscript critical comments by authors are extremely brief and rare to find, summaries with tables after each chapter are not enough to cover all presented data. Presented studies need to be discussed more in detail, preferably by strengths, weaknesses, and opportunities - following the proposed title.

Introduction (line 45) - It would be beneficial for the text to insert data from other countries about consumer preferences. 

Paragraph - Absence of hunger - no data regarding feed palatability and it's effect on welfare of pigs.

Paragraph - Absence of thirst has no scientific data which could even be commented on. Paragraph about how to install a water pipe in-ground is not scientific review. If there are no research studies regarding effect of thirst it should be pointed out and commented on that. 

Line 135-137 definition is lacking resting as one of the key issues in the comfort of animals. Also, a little data about the same issue, but in Table 1. there are scores for each category, although data were not presented.

3.1. Bio-exclusion line 305-312- all presented data regarding the infectious diseases are basicly thosewho emerged in conventional systems over the last decades, and yet authors in the table 2 score +++, again without any discussion. Line 358-359 - what about other countries?

Line 402-403Reference for this statement?

Line 694-695 This statement needs to be discussed. It is well known that all rearing systems have advantages and disadvantages but here the authors need to discuss them.

Line 733-734 - Authors should check more in-depth data as google scholar show about 10700 results on "Yersinia Prevalence In pigs in different rearing systems". If authors have stated four major bacteria that are interesting for meat safety all four should be looked more in-depth, not only one. 

Line 894 (Summary) - no discussion about microbiological and parasitological risks whatsoever. 

Conclusions are written in a way of a short discussion. They need to be revised completely. 

Reviewer 2 Report

I really enjoyed reading this manuscript. The topic is very important and it is crucial to objectively show the whole picture. Authors did this perfectly.

The public is always concerned about animal welfare, mainly the freedom of natural behaviour but the very important questions of diseases, zoonoses and antibiotic resistance are often overlooked. Thank you for involving these issues in the manuscript.

My only concern is about the Tables. They are hard to read, maybe due to the layout of the first column. This should not be aligned to the centre, rather to left. Otherways, I liked the tables, they summarise well the antecedent paragraphs.

I miss one thing from the manuscript: Could you please summarise the knowledge about the weaning (methods, the effects on the behaviour and health of the piglets) in the different systems? This is also an important animal welfare issue.

Another idea that may be in the manuscript: Describe the economic background of the different systems. Why the farmers choose one or another? Are there clear economic advantages or disadvantages of them? I think that this could be the last section of the manuscript. Or, if this is too long or complicated, this is an idea for your next review.

Round 2

Reviewer 1 Report

I would like to thank the authors for taking into consideration all of the recommendations, which were sent in good faith. After significant revision, the manuscript is now comprehensive, easy to read, and very informative. Critical comments from authors gave it the "depth" needed to be classified as a scientific review. 

There is still something to change and that is line 92 - the complete number of papers reviewed. As the authors added several new literature sources, the total number of papers is now 223 rather than 198 as it is stated. 

Author Response

Point 1: I would like to thank the authors for taking into consideration all of the recommendations, which were sent in good faith. After significant revision, the manuscript is now comprehensive, easy to read, and very informative. Critical comments from authors gave it the "depth" needed to be classified as a scientific review. 

Response 1: We are pleased to read your comment and thank you for your positive feedback.

Point 2: There is still something to change and that is line 92 - the complete number of papers reviewed. As the authors added several new literature sources, the total number of papers is now 223 rather than 198 as it is stated.

Response 2:  You are right, thank you for this comment. The total number of references is indeed 223, including references from our personal archives in addition to those of the search engines PubMed and Google Scholar. We have modified, as suggested, the number of references but have also revised the following sentence to clarify this point (line 92).
